# Analysis of Cadmium Accumulation Characteristics Affected by Rhizosphere Bacterial Community of Two High-Quality Rice Varieties

**DOI:** 10.3390/plants14121790

**Published:** 2025-06-11

**Authors:** Shangdu Zhang, Zhengliang Luo, Ju Peng, Xiang Wu, Xiufei Meng, Yuanyi Qin, Feiying Zhu

**Affiliations:** 1Guizhou Academy of Agricultural Sciences, Guiyang 550006, China; pengju654321@126.com (J.P.); gzgyrice@163.com (X.W.); 18798023791@163.com (X.M.); qyy_16@163.com (Y.Q.); 2Hunan Academy of Agricultural Sciences, Changsha 410125, China; luozl@hunaas.cn

**Keywords:** cadmium, microbiome, rice, rhizosphere bacterial

## Abstract

Cadmium-contaminated rice poses serious health risks through the bioaccumulation of Cd (cadmium) from soil to edible grains. Cd contamination disrupts soil microbial ecology and alters microbial diversity. However, the role of cultivar-specific rhizosphere microbial communities in modulating Cd uptake remains unclear. In this study, we aimed to elucidate the mechanism underlying variety-dependent rhizosphere microecological remodeling and Cd accumulation in two independently selected late rice varieties, Yuzhenxiang (YZX) and Xiangwanxian 12 (XWX12). Combining physiological and metagenomic analyses, we revealed variety-specific correlations between root Cd accumulation and dynamic changes in soil pH, soil available phosphorus, and rhizosphere bacteria. The key bacterial genera (*Variibacter*, *Nitrospira*) showed differential enrichment patterns under Cd stress. In contrast, *Galella* and *Anaeromyxobacter* likely reduce Cd bioavailability by modulating phosphorus availability. Overall, this study elucidates that rice cultivars indirectly shape Cd accumulation patterns via rhizosphere microbial remodeling, providing novel insights for microbial remediation strategies in Cd-contaminated farmland.

## 1. Introduction

Rice (*Oryza sativa* L.) serves as the primary staple food for approximately half of the global population [1,2]. However, the agricultural sector faces mounting challenges in meeting the demands of an increasing population due to limited suitable cultivation land. In recent years, significant portions of rice crops have been cultivated in contaminated soils due to various anthropogenic and environmental factors [3]. Cadmium (Cd) is a common heavy metal contaminant in rice cultivation [4,5]. Excessive Cd uptake and accumulation can hinder rice growth and reduce rice grain quality [6]. Additionally, the consumption of rice contaminated with Cd can trigger multiple health problems, such as cancer and cardiovascular, reproductive, and nervous system diseases, in humans [3].

Previous studies have recently demonstrated that Cd contamination significantly disrupts soil microbial ecology, altering both microbial diversity and community structure [7,8,9]. Recent investigations have focused on elucidating the mechanisms governing Cd accumulation in the rhizosphere environment. Notably, functional rhizosphere microbes play crucial roles in cell wall retention of Cd, secretion of organic acids that modify soil pH, and mediating Cd bioavailability and accumulation. In response to Cd stress, plants employ adaptive strategies by releasing specific root exudates, including long-chain fatty acids, amino acids, short-chain organic acids, and sugars. These exudates facilitate the recruitment of Cd-responsive rhizosphere microbes, such as *Shewanella putrefaciens*, *Bacillus megatherium*, and *Pseudomonas aeruginosa*. A growing body of evidence confirms that such microorganisms can establish symbiotic relationships with host plants, thereby enhancing the stability and activity of microbial communities involved in Cd absorption and transport mechanisms in crops [10,11].

Cd accumulation in rice plants and grains is influenced by multiple factors, including soil physicochemical properties, rhizosphere microorganisms, and rice plant genotypes [12,13]. Cd contamination significantly alters soil microbial ecology by reducing diversity and restructuring communities [4,7,14]. For instance, Song et al. [7] demonstrated that Cd-polluted soils exhibit a significant decrease in microbial richness, community evenness, and community structures. Furthermore, rhizosphere bacterial communities show genotype-dependent responses to Cd stress. Li et al. [6] reported differential microbial colonization between transgenic and wild-type rice varieties and varietal-specific microbial responses to Cd exposure. Current remediation strategies for Cd-contaminated paddy fields include water management, fertilization, soil removal and replacement, chemical remediation, and organic amendments (e.g., biochar, composts, and manures) [6,15]. Microbial remediation has emerged as a particularly promising strategy due to its high efficiency, economic benefits, environmental compatibility, eco-friendliness, and safety relative to conventional techniques [16,17]. Accumulating microorganisms have been implicated in the bioremediation of Cd-contaminated farmland and the reduction of Cd accumulation in rice plants [18,19,20]. Key mechanisms of microbial Cd mitigation include biomineralization, Cd precipitation, biosorption, cell wall binding, biotransformation, valence modification, bioaccumulation, and intracellular sequestration [21]. Notably, specific bacterial phyla demonstrate Cd mitigation potential, including Proteobacteria for Cd immobilization, Firmicutes for plant growth promotion, and Bacteroidetes for Cd bioavailability reduction [14].

Cadmium-contaminated rice, characterized by excessive cadmium accumulation from soil to grains and other plant organs, poses significant risks to human health. Our previous research demonstrated substantial differences in cadmium accumulation patterns between two representative rice varieties: Yuzhenxiang (YZX) and Xiangwanxian 12 (XWX12) [9,22]. Building upon these findings, the present study investigates the influence of rhizosphere bacterial communities on cadmium accumulation in these two rice varieties during their mature growth stage under three distinct cadmium stress conditions.

## 2. Results

### 2.1. Differences in Phenotypes and Cd Contents Between YZX and XWX12 Under Different Cd Stress Conditions

A comparison of the phenotypes of YZX and XWX12 at the mature stage is shown in Figure 1A,B. YZX had an average plant height of 117 cm, a grain length of 0.95 cm, and a length-to-width ratio of 4.8, while XWX12 had an average plant height of 96 cm, a grain length of 0.76 cm, and a length-to-width ratio of 3.6. Given the variations in the absorption, translocation, and accumulation of heavy metals among plant organs and cultivars, the Cd content in different organs of these two rice strains (YZX and XWX12) was examined at the A, B, and C cultivation sites (Figure 1C,D). The results showed that the Cd content in roots tended to increase with increasing soil Cd content (Figure 1E,F). In addition, these two rice cultivars exhibited diverse cadmium (Cd) contents in the same organ and cultivation base (Figure 1). The results demonstrated that the root Cd content was greater than that in the stems, leaves, or grains of both rice cultivars.

### 2.2. The Dynamic Changes in Soil Physicochemical Properties Under Different Cd Contents

There was a significant difference in the soil physicochemical properties between the A/B and C groups, such as pH, soil organic matter (SOM), soil total nitrogen (TN), soil total phosphorus (TP), soil available nitrogen (AN), soil available phosphorus (AP), and soil available potassium (AK) (Figure 2A,C). The results showed significant differences in soil pH, TN, TP, and AP among the six sample groups (Figure 2D,G). Notably, the differences in soil pH and AP under moderate-Cd stress conditions can be attributed to disparities in Cd content in rice roots.

### 2.3. Altered Rhizosphere Bacterial Community Structure Under Increasing Cd Stress Conditions in YZX and XWX12

The average number of reads per sample was 56,145. We conducted a diversity analysis of bacteria at the operational taxonomic unit (OTU) level in all samples (Figure 3A–C). The rhizosphere bacterial community richness and diversity differed significantly between YZX and XWX12 (Figure 3D,E). A Venn diagram revealed 2602 overlapping bacterial OTUs (Figure 4A), 43 overlapping bacterial phyla (Figure 4B), and 607 overlapping bacterial genera (Figure 4C) across all samples. Notably, the abundances of OTU 7161 (P_WA-aaa01f12) and OTU1023 (P_SR1_Absconditabacteria) declined significantly in the B group (Figure 4D). Community bar plot analysis showed that the most abundant bacterial phyla were Proteobacteria, Chloroflexi, Actinobacteria, Acidobacteria, Firmicutes, and Nitrospirae (Figure 4E).

To determine significant differences in rhizosphere bacterial communities under three cadmium (Cd) stress conditions, we performed Kruskal–Wallis H tests to identify the top 30 phyla and genera in the rhizosphere bacterial community (Figure 5). Under low-Cd stress conditions (A), the dominant phyla in YZX were Actinobacteria, Acidobacteria, Nitrospirae, Cyanobacteria, and Bacteroidetes, while the dominant phyla in XWX12 were Proteobacteria, Actinobacteria, Acidobacteria, and Nitrospirae. Under medium-Cd stress conditions (B), the dominant phyla in YZX were Acidobacteria, Actinobacteria, Nitrospirae, Bacteroidetes, and Cyanobacteria, while the dominant phyla in XWX12 were Proteobacteria, Actinobacteria, Acidobacteria, and Nitrospirae. Under high-Cd stress conditions (C), the dominant phyla of YZX were Actinobacteria, Acidobacteria, Nitrospirae, Cyanobacteria, and Bacteroidetes, while the dominant phyla of XWX12 were Proteobacteria, Actinobacteria, Acidobacteria, and Planctomycetes (Figure 5A,B). The Actinobacteria and Acidobacteria abundances declined significantly under medium-Cd stress conditions in both YZX and XWX12. Under low-Cd conditions, the dominant genera in YZX were *Nitrospira*, *Anaeromyxobacter*, *Oryzihumus*, *Variibacter*, and *Bradyrhizobium*, while the dominant genera in XWX12 were *Nitrospira*, *Anaeromyxobacter*, *Geobacter*, *Oryzihumus*, and *Variibacter*. Under medium-Cd stress conditions (B), the dominant genera in YZX were *Nitrospira*, *Anaeromyxobacter*, *Desulfobacca*, *Anaeromyxobacter*, and *Bradyrhizobium*, while the dominant genera in XWX12 were *Nitrospira*, *Variibacter*, *Geobacter*, *Anaeromyxobacter*, and *Bradyrhizobium*. Under high-Cd stress conditions, the dominant genera in YZX were *Nitrospira*, *Variibacter*, *Acidothermus*, *Bradyrhizobium*, and *Anaeromyxobacter*, while the dominant genera in XWX12 were *Nitrospira*, *Variibacter*, *Acidothermus*, *Bradyrhizobium*, and *Anaerolinea* (Figure 5C,D).

Notably, our results indicated that the genus *Nitrospira*’s abundance increased similarly under medium-Cd stress in both cultivars. Medium-Cd stress emerged as the critical threshold for altering rhizosphere bacterial community structure.

We employed the Wilcoxon rank-sum test to identify significant differences in rhizosphere bacterial communities between the two rice varieties (Figure 6). At the phylum level, Cyanobacteria was significantly more abundant in YZX than in XWX12 under low-Cd stress conditions (Figure 6A). Significant abundance differences were observed for Proteobacteria, Chloroflexi, Actinobacteria, Nitrospirae, Firmicutes, and Bacteroidetes under moderate-Cd stress conditions (Figure 6B). Bacteroidetes, Nitrospirae, and Caldiserica showed significant abundance variations under high-Cd stress conditions (Figure 6C). Furthermore, at the genus level, *Roseiflexus*, *Leptolyngbya*, *Chloronema*, *Pedomicrobium, Syntrophus*, *Arthronema*, *Gemmatirosa*, *Methylomonas*, *Leptonema*, and *Anacrosporobacter* were significantly enriched in YZX under low-Cd stress conditions (Figure 6D). *Nitrospira*, *Desulfobacca*, *Mycobacterium*, *Variibacter*, *Gaiella*, *Phormidium*, *Synechocystis*, and *Rhodoplanes* exhibited significant abundance differences under moderate-Cd stress conditions (Figure 6E). *Geobacter*, *Rhodanobacter*, *Arlsobacter*, *Nocardioides*, *Roseomonas*, *Deferrisoma*, *Aquicella*, *Paludibaculum*, *Desulfobulbus*, and *Chitinophaga* showed significant variations under high-Cd stress conditions (Figure 6F). The Spearman correlation analysis of Cd accumulation in rice roots showed that the phyla of Chloroflexi (r = 0.85, *p* < 0.01) and Nitrospirae (r = 0.82, *p* < 0.01) positively correlated with root Cd accumulation, while Proteobacteria (r = −0.76, *p* < 0.05), Firmicutes (r = −0.88, *p* < 0.001), Bacteroidetes (r = −0.88, *p* < 0.001), and Cyanobacteria (r = −0.67, *p* < 0.01) correlated at the phylum level. Furthermore, negative correlations were observed for the genera of *Anaeromyxobacter* (r = −0.75, *p* < 0.05), *Gaiella* (r = −0.77, *p* < 0.01), *Clostridium_sensu_stricto_1* (r = −0.79, *p* < 0.01), *Defluviicoccus* (r = −0.88, *p* < 0.01), *Mycobacterium* (r = −0.88, *p* < 0.001), and *Desulfobacca* (r = −0.85, *p* < 0.01). Positive correlations were found for *Nitrospira* (r = 0.82, *p* < 0.01), *Bryobacter* (r = 0.73, *p* < 0.05), *Bradyrhizobium* (r = 0.68, *p* < 0.05), *Variibacter* (r = 0.63, *p* < 0.05), and *Thiobacillus* (r = 0.56, *p* < 0.01) (Figure 7B).

Additionally, PICRUSt2-based analysis revealed that the most abundant bacterial functions were associated with amino acid metabolism, translation, ribosomal structure, biogenesis, energy production and conversion, and cell wall/membrane/envelope biogenesis (Figure 7C). The key metabolic pathways included phenylalanine biosynthesis, porphyrin and chlorophyll metabolism, propanoate metabolism, homologous recombination, alanine, aspartate, glutamate metabolism, and the TCA cycle (Figure 7D).

## 3. Discussion

### 3.1. How Rice Varieties and Different Cd Stress Conditions Influenced the Cd Content

Numerous studies have documented significant interspecific variations in Cd accumulation patterns among crops, particularly in vegetables across diverse geographical regions. The root system serves as the primary site for Cd uptake, with well-documented evidence demonstrating significantly higher Cd accumulation in root tissues than in aerial plant parts [23,24]. Root exudates comprise diverse components, including inorganic ions, protons, and organic compounds. These exudates are released from different segments of the root system during plant growth and are an inherent physiological characteristic of roots [25]. Consistently, our results demonstrated significantly higher root Cd concentrations in root tissues compared to stems, leaves, or grains of both rice cultivars (*p* < 0.5), underscoring the critical role of root systems in manipulating Cd accumulation.

Furthermore, Cd accumulation in rice plants exhibits multifactorial dependence, being modulated by soil physicochemical properties, rhizosphere microorganisms, and rice plant genotypes [12,13]. Our study further confirmed that rice roots are key organs influencing the Cd content in plants under different soil Cd stress levels between YZX and XWX 12 (Figure 1). Importantly, our results exhibited the same trend of Cd enrichment, which increased with increasing soil Cd concentration, between these two different rice cultivars, indicating that soil Cd stress plays an essential role in influencing rice Cd accumulation. Alterations in soil physicochemical properties significantly influence rhizosphere microbial community composition (Figure 2).

### 3.2. The Rhizosphere Bacterial Community Has a Relationship to Rice Cd Accumulation

Under moderate-Cd stress, soil pH and available phosphorus (AP) exhibited significant alterations (Figure 2A–C), showing a strong correlation with root Cd accumulation. Spearman correlation analysis was employed to elucidate the relationships between environmental factors and microbial community dynamics. Proteobacteria have been shown to acidify soil pH and enhance Cd bioavailability through organic acid secretion [23]. Our results confirmed that the Proteobacteria, Firmicutes, and Bacteroidetes promote plant growth while reducing Cd accumulation in rice roots (Figure 7A). However, Chloroflexi abundance showed an inverse relationship with Cd accumulation, potentially through cell wall adsorption or extracellular polymeric substances (EPS)-mediated chelation [18]. Research on the function of Chloroflexi in the evolution of photosynthesis, which involves the fixation of inorganic CO_2_ and aerobic oxidation of carbon and nitrite, has been reported [26,27]. PICRUST2 functional prediction analysis revealed that Chloroflexi and Nitrospirae regulate the carbon–nitrogen cycle (e.g., CO_2_ fixation and nitrite oxidation), which may modify Cd speciation (e.g., promoting Cd^2+^ dissolution) and subsequent plant uptake [28,29] (Figure 7C).

Cd pollution has been widely documented to induce oxidative stress, which directly mediates Cd toxicity in plants [30,31]. Oxygen availability significantly influences Cd accumulation dynamics in rice [32,33]. Enhanced populations of antioxidant bacteria and upregulated defense responses may confer Cd tolerance in rice [32,34,35]. Additionally, mobile elements (C, O, N), facilitate bacterial adaptation and heavy metal resistance through soil biogeochemical cycling [33,36]. At the genus level, *Variibacter* and *Nitrospira* exhibited significantly differential enrichment patterns between YZX and XWX12 cultivars under Cd stress (Figure 7B). PICRUST2 analysis predicted *Variibacter* involvement in organic acid metabolism (COG category: carbohydrate transport and metabolism) (Figure 7C). Conversely, *Gaiella*, *Mycobacterium* [37], Desulfobacca [38], and *Anaeromyxobacter* reduced Cd bioavailability via phosphorus activation, thereby decreasing accumulation (Figure 7B–D). Fe-reduction-related oxidative phosphorylation is driven by *Anaeromyxobacter* [38] and *Gaiella*’s potential role in prokaryotic carbon fixation pathways [39]. These findings demonstrate that the rice genotype modulates Cd accumulation through rhizosphere microbial community regulation [40].

Accumulating evidence indicates dynamic compositional and functional shifts in rhizosphere microbiota throughout the rice growth cycle. These temporal microbial dynamics mediate Cd bioavailability through multifaceted mechanisms. Collectively, our results identified four rhizobacterial genera (*Variibacter, Nitrospira*, *Galella*, and *Anaeromyxobacter*) that critically regulate root Cd accumulation during maturation, which may directly influence observed accumulation patterns.

The environmental effect, genotype by environment interactions, and random variability in the rice rhizosphere bacterial community are usually low across greenhouse experiments under controlled conditions [25,35]. However, there can be noticeable variation in the rhizosphere bacterial community across different environments. Therefore, the present findings need to be reassessed under multiple field conditions before a definitive claim regarding the real difference between the studied genotypes can be made.

## 4. Materials and Methods

### 4.1. Experimental Design

The experiment was carried out at the Chunhua Scientific Research Base of the Hunan Academy of Agricultural Sciences (Changsha, China) in an artificial intelligence controlled greenhouse (113°26′57.2° E, 28°29′36.9° N). We selected Yuzhenxiang (YZX, 1) and Xiangwanxian 12 (XWX 12, 2) as experimental materials based on our previous research [9,22]. Three uniform experimental units were planted in an area with a self-closing steel greenhouse and a set of ponds for irrigation and drainage, which were controlled by remote control system equipment to maintain the consistency of environmental factors. The soil for planting contained natural cadmium content treatments, which were conducted in three experimental unit cultivation identification areas with total cadmium contents of 0.16 ± 0.1 mg/kg (Low Cd_1, A), 0.5 ± 0.1 mg/kg (Medium Cd_2, B), and 0.9 ± 0.1 mg/kg (High Cd_3, C). The rice seeds were sown on 13 June 2022, and the plants were subsequently transplanted on 1 July 2022. The experiment was arranged in a completely randomized block design with three replications. Each cultivar was grown in two rows of 14 hills per row, with two seedlings transplanted per hill. Plant spacing was 17 cm × 20 cm (within rows × between rows). Each experimental unit comprised 84 seedlings in total. Fertilizer management included no base fertilizer, urea (60 kg/ha), or potassium chloride (30 kg/ha) applied to all three ponds. Water management included intermittent irrigation with three controlled drought periods during the growth cycle: (i) the peak tillering stage (3-day drought), (ii) the post-heading stage (3-day drought), and (iii) the pre-harvest stage (3-day drought). Continuous flooding was maintained during intervening periods. Irrigation was resumed upon visible initiation of soil surface cracking to establish standardized moisture stress for phenotypic selection. After the removal of foreign matter, such as impurities, stones, and plant residues, we used a spoon to scrape the rhizosphere soil from the rice roots. Our experiments were divided into 6 groups (A_1, A_2, B_1, B_2, C_1, and C_2). Each group contained 5 biological replicates from each experimental unit. A total of 30 rhizosphere soil samples were collected from two rice varieties under three different Cd stress conditions.

### 4.2. Determination of Cd Content in Plants

The plant samples were ground repetition per repetition using a grinder (the machine was cleaned first to avoid cross-contamination between the next material and the previous one), passed through a sieve, and stored in a dryer. Dry samples (0.5000 g) were weighed and placed in a microwave digestion tank, and nitric acid (5 mL) and hydrogen peroxide (2 mL) were added. After digestion, the acid was heated until nearly dry, the digestion tank was rinsed with nitric acid solution three times, the solution was transferred to a 25 mL volumetric flask, the solution was diluted to scale with nitric acid solution, and the solution was mixed well. Moreover, reagent blank tests were performed. Atomic absorption spectrophotometry for the determination of cadmium content was performed according to the instructions of the standard “National Food Safety Standard for the Determination of Cadmium in Foods“ (GB 5009.15-2014) [41].

### 4.3. Soil Physicochemical Property Determination

The soil properties and the cadmium content in the plant roots, stem sheaths, leaves, and grains at the rice maturity stage were tested via the microwave digestion method. After the removal of foreign matter, such as impurities, stones, and plant residues, 10 soil samples under each treatment condition were mixed, air-dried, pulverized, and filtered through 10-mesh standard sieves (2 mm sieves). Next, the physicochemical properties were determined. ① The total cadmium content in the soil was measured through the “complete digestion method” using graphite furnace atomic absorption spectrophotometry (Soil Quality—Determination of lead, cadmium-Graphite furnace atomic absorption spectrophotometry (GB/T 17141-1997) [42]). ② The effective cadmium content in the soil samples was examined through the “leaching method” using graphite furnace atomic absorption spectrophotometry (Soil quality—Analysis of available lead and cadmium contents in soils—Atomic absorption spectrometry (GB/T 23739-2009) [43]). ③ The total nitrogen and alkaline soluble nitrogen in the soil samples were determined using the Kjeldahl and alkaline hydrolysis diffusion methods (Nitrogen determination methods of forest soils (LY/T1228-2015) [44]), respectively. ④ The perchloric acid–sulfuric acid method (Method for determination of soil total phosphorus (GB9837-88) [45]) was used to determine the total phosphorus content. ⑤ The available phosphorus content was determined using a spectrophotometer (Soil testing-Part 7: Method for determination of available phosphorus in soil (NY/T 1121.7-2014) [46]). ⑥ The total potassium content was measured using the sodium hydroxide melting method (Method for determination of soil total phosphorus (GB 9837-88) [45]). ⑦ The available potassium was measured through flame atomic absorption spectrophotometry (Determination of exchangeable potassium and non-exchangeable potassium content in soil (NY/T889-2004) [47]). ⑧ The pH was determined using a redox potentiometric at a water–soil–liquid ratio of 1:2.5 (Soil Testing Part 2: Method for determination of soil PH (NY/T 1121.2-2006) [48]). ⑨ The soil organic matter content was measured through redox calorimetry (Soil Testing Part 2: Method for determination of soil organic matter (NY/T 1121.6-2006) [49]). 

### 4.4. Soil DNA Extraction and 16S rDNA Sequencing 

Our experiments were divided into 6 groups (i.e., A_1, A_2, B_1, B_2, C_1, and C_2). Each group contained 5 biological replicates. A total of 30 soil samples were collected from two rice varieties under three different Cd stress conditions. After the removal of foreign matter, such as impurities, stones, and plant residues, we used a spoon to scrape the rhizosphere soil from the rice roots. The E.Z.N.A ^®^ soil DNA kit (Omega Biotek, Norcross, GA, USA) was used. Firstly, we briefly centrifuged the E-Z 96 Disruptor Plate C Plus to remove any ceramic beads from the walls of the wells. We uncapped the E-Z 96 Disruptor Plate C Plus and saved the caps for use. After adding 10–250 mg of soil sample, we added 525 µL of SLX-Mlus Buffer and 2 µL of RNase A and sealed the plate with the caps removed in step 1. Then, we followed the manufacturer’s protocols to obtain the purified DNA. The DNA content and purity were analyzed using a NanoDrop 2000 UV-vis spectrophotometer (Thermo Scientific, Wilmington, NC, USA). DNA samples were stored at −80 °C until sequencing. The V3–V4 hypervariable regions of the bacterial 16S rRNA genes were amplified using the above DNA templates of the primers 338F (5′-ACTCCTACGGGAGGCAGCAG-3′) and 806R (5′-GGACTACHVGGGTWTCTAAT-3′) on an ABI GeneAmp 9700 PCR thermocycler (ABI, CA, USA) under the following PCR procedures: 95 °C for 3 min, 27 cycles of 95 °C for 30 s, 55 °C for 30 s, and 72 °C for 45 s, and 72 °C for 10 min. The PCR products were retrieved after 2% agarose gel electrophoresis and purified using an AxyPrep DNA Gel Extraction Kit (Axygen Biosciences, Union City, CA, USA) following the manufacturer’s protocols and quantified using a Quantus Fluorometer (Promega, Madison, WI, USA). The 16S rDNA library was constructed using the NEXTFLEX Rapid DNA-Seq Kit (Bio Scientific, Austin, TX, USA) according to the manufacturer’s instructions and then paired-end sequenced on the Illumina MiSeq PE300 platform (Majorbio Bio-Pharm Technology Co., Ltd. Shanghai, China). The raw data were deposited into the NCBI Sequence Read Archive (SRA) database (accession no. PRJNA602547).

### 4.5. Sequencing Data Processing

The raw sequencing reads were filtered and processed using fastp (version 0.20.0) and FLASH (version 1.2.7) software under the following conditions: (i) reads with an average quality score of <20, a length shorter than 50 bp, and ambiguous characters were removed; (ii) only overlapping sequences longer than 10 bp were assembled, and reads that could not be assembled were removed. The maximum mismatch ratio of the overlap region was 0.2. Operational taxonomic units (OTUs) with a 97% similarity cutoff were clustered, and chimeric sequences were discarded using UPARSE version 7.1 software. Each representative OTU sequence was classified and annotated. The taxonomy of each representative OTU sequence was compared against the 16S rRNA database (Silva v138) under a confidence threshold of 0.7 using RDP Classifier version 2.2. Bioinformatic analysis of the soil/gut microbiota was carried out using the Majorbio Cloud platform (https://cloud.majorbio.com, accessed on 25 June 2024). Based on the OTU information, rarefaction curves and alpha diversity indices, including observed OTUs, Chao1 richness, Shannon index, and Good’s coverage, were calculated with Mothur v1.30.1 [50]. The similarity among the microbial communities in different samples was determined through principal coordinate analysis (PCoA) based on Bray–Curtis dissimilarity using the Vegan v2.5-3 package. A correlation between two nodes was considered to be statistically robust if Spearman’s correlation coefficient was over 0.6 or less than −0.6 and the *p*-value was less than 0.01.

### 4.6. Functional Trait Prediction of the Rhizosphere Bacterial Community

The metagenomic function was predicted using PICRUSt2 (Phylogenetic Investigation of Communities by Reconstruction of Unobserved States) [29] based on OTU representative sequences.

### 4.7. Statistical Analysis

Statistical data analysis was performed using GraphPad Prism version 7 (La Jolla, CA, USA). The results are displayed as the means ± standard deviations.

## 5. Conclusions

In this study, our results indicated that the key bacterial genera (*Variibacter*, *Nitrospira*) exhibited significant differences in enrichment under cadmium stress. In contrast, *Galella* and *Anaeromyxobacter* likely reduce Cd bioavailability by modulating phosphorus availability. These findings not only elucidate the differential mechanism of Cd accumulation in YZX and XWX12 from the perspective of microbial ecology but also establish a theoretical basis for developing microbial remediation strategies in Cd-contaminated farmland. Overall, our experiment elucidates that rice cultivars indirectly shape Cd accumulation patterns via rhizosphere microbial remodeling, providing novel insights for microbial remediation strategies in Cd-contaminated farmland.

## Figures and Tables

**Figure 1 plants-14-01790-f001:**
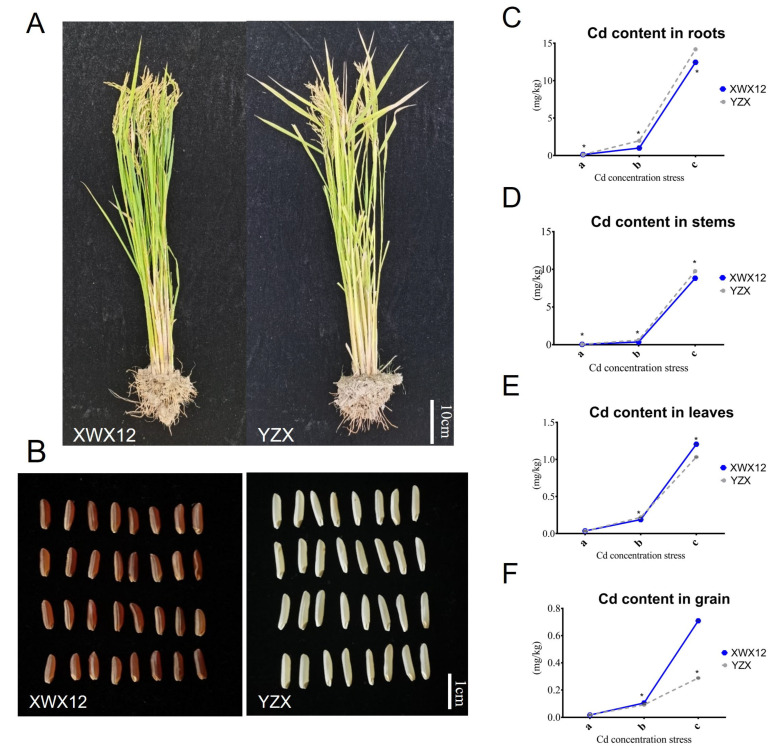
Phenotypes and Cd content in plant organs upon comparison of the two tested rice varieties at the mature stage. Phenotypic comparison of the two tested rice varieties at the mature stage (**A**,**B**). Comparative analysis of the Cd contents of roots (**C**), stems (**D**), leaves (**E**), and grains (**F**) of 2 rice cultivars under different Cd stress conditions. Note: a (Low Cd_1, a), b (Medium Cd_2, b), c (High Cd_3, c), *t* test * *p* < 0.05.

**Figure 2 plants-14-01790-f002:**
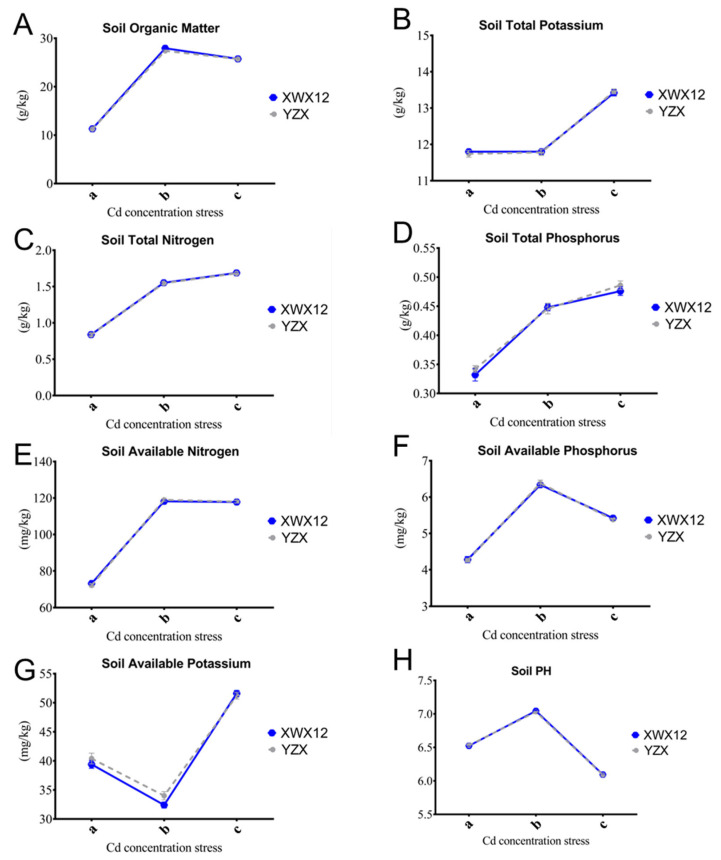
Comparative analysis of soil chemical properties. Bar plot of significantly different soil physicochemical properties between the a (Low Cd_1, a)/b (Medium Cd_2, b) Cd content (**A**), a (Low Cd_1, a)/b (High Cd_3, c) Cd content (**B**), and b (Medium Cd_2, b)/c (High Cd_3, c) Cd content (**C**) treatments (*p* < 0.0001). (**A**). Comparison of soil organic matter. (**B**) Comparison analysis of soil total potassium. (**C**) Comparison analysis of soil total nitrogen. (**D**) Comparison analysis of soil total phosphorus. (**E**) Comparison analysis of soil AP. (**F**) Comparison analysis of soil available phosphorus. (**G**) Comparison analysis of soil available potassium. (**H**) Comparison analysis of soil PH.

**Figure 3 plants-14-01790-f003:**
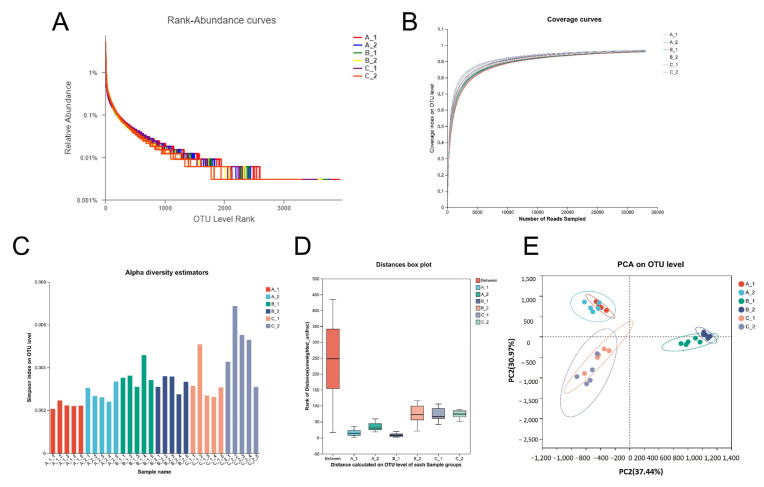
Diversity analysis of bacteria at the operational taxonomic unit (OTU) level in all samples. (**A**) Rank abundance curves. (**B**) Alpha diversity analysis of the coverage curves. (**C**) Alpha diversity analysis of the Simpson index. (**D**) Distances box plot of beta diversity. (**E**) PCA at the OTU level. Note: A_1 (Low Cd_1, YZX), A_2 (Low Cd_2, XWX12), B_1 (Medium Cd_2, YZX), B_2 (Medium Cd_2, XWX12), C_1 (High Cd_3, YZX), and C_2 (High Cd_3, XWX12).

**Figure 4 plants-14-01790-f004:**
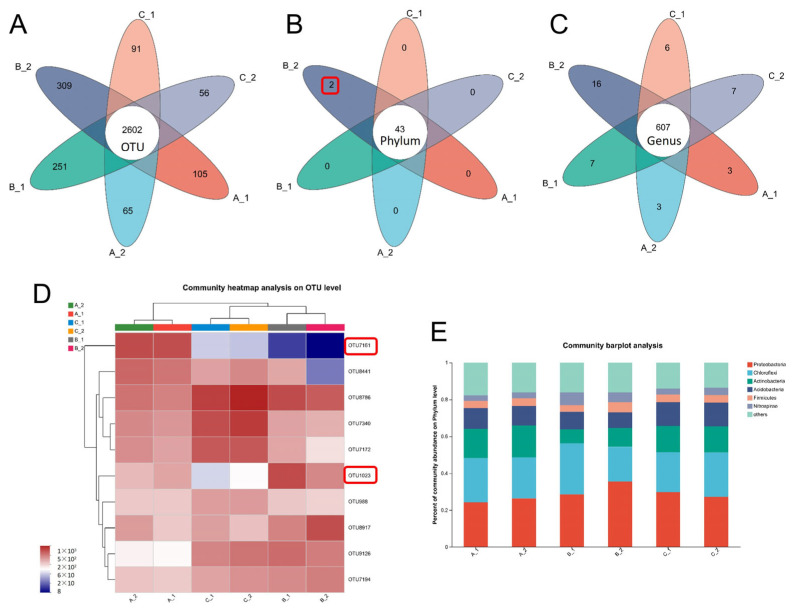
Rhizosphere bacterial community composition analysis. Venn diagram of bacterial community composition between different groups at the OTU (**A**), phylum (**B**), and genus (**C**) levels. (**D**) Community heatmap analysis of different groups at the OTU level. (**E**) Bacterial community bar plot at the phylum level. Note: A_1 (Low Cd_1, YZX), A_2 (Low Cd_2, XWX12), B_1 (Medium Cd_2, YZX), B_2 (Medium Cd_2, XWX12), C_1 (High Cd_3, YZX), and C_2 (High Cd_3, XWX12).

**Figure 5 plants-14-01790-f005:**
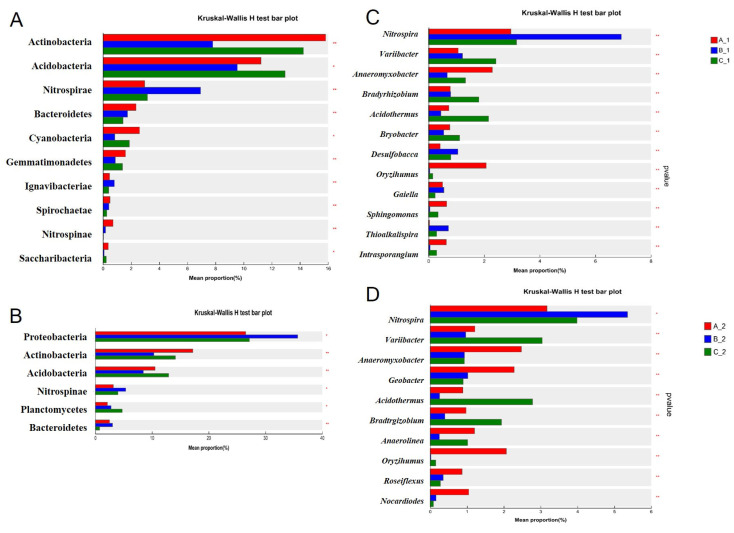
Statistical differences in rhizosphere bacterial communities under different Cd stress conditions. The top 30 significantly different phyla (**A**) and genera (**B**) in YZX (1) were identified through the Kruskal-Wallis H test under different Cd stress conditions. The top 30 significantly different phyla (**C**) and genera (**D**) in XWX12 (2) were identified through the Kruskal-Wallis H test under different Cd stress conditions. Note: * *p* < 0.05, ** *p* < 0.001.

**Figure 6 plants-14-01790-f006:**
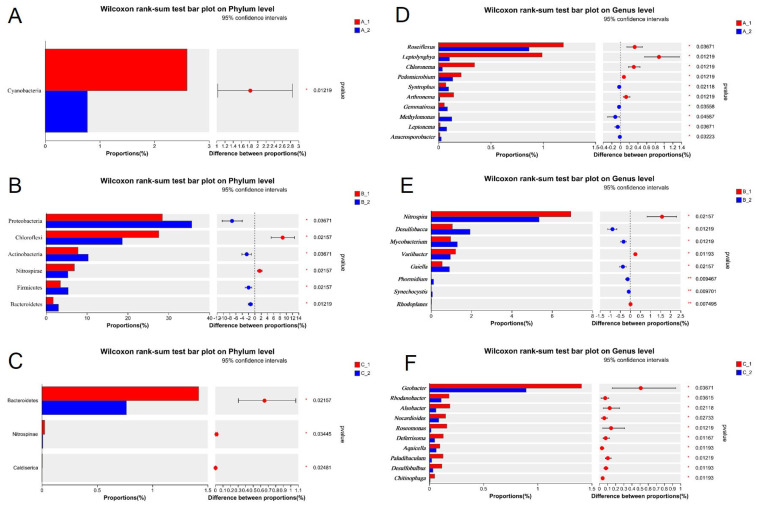
Statistical differences in the rhizosphere bacterial community between the two rice varieties. Significant differences in the rhizosphere bacterial community between YZX (1) and XWX12 (2) under low-Cd conditions at the phylum (**A**) and genus (**B**) levels. Significant differences in the rhizosphere bacterial community between YZX and XWX12 under moderate-Cd stress conditions at the phylum (**C**) and genus (**D**) levels. Significant differences in the rhizosphere bacterial community between YZX and XWX12 under high-Cd stress conditions at the phylum (**E**) and genus (**F**) levels. Note: * *p* < 0.05, ** *p* < 0.001.

**Figure 7 plants-14-01790-f007:**
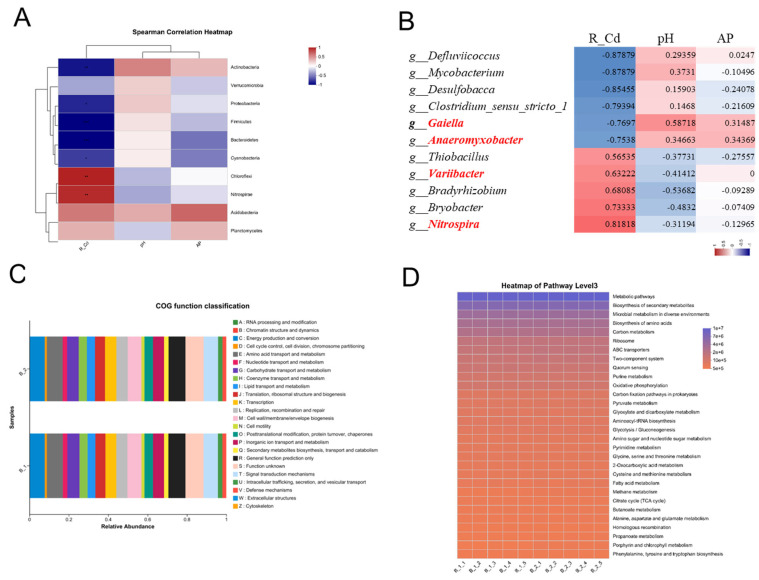
Spearman correlation heatmap of environmental factors and prediction analysis of bacterial functions. (**A**) Spearman correlation heatmap for environmental factors and the top 10 phyla under medium-Cd stress conditions. (**B**) Spearman correlation heatmap for environmental factors and the top 30 genera under medium-Cd stress conditions. (**C**) COG functional classification of bacteria. (**D**) Heatmap of bacterial functional KEGG pathways. Note: The correlation coefficient R is indicated by different colors. * *p* value < 0.05, ** *p* value < 0.01, *** *p* value < 0.001.

## Data Availability

The clean reads were deposited into the NCBI Sequence Read Archive (SRA) database (accession number: SUB11813545).

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
