# Peer review of "Analysis of Cadmium Accumulation Characteristics Affected by Rhizosphere Bacterial Community of Two High-Quality Rice Varieties"

_plants, 2025, doi:10.3390/plants14121790_

Round 1

Reviewer 1 Report

Comments and Suggestions for Authors

The manuscript entitled “Analysis on Cadmium Accumulation Characteristics effected by Rhizosphere Bacterial Community of Two Self-selected High-Quality Rice” has been reviewed in detail. The researchers employed an integrated approach combining physiological measurements and metagenomic analysis to characterize the rhizosphere bacterial communities under three different Cd stress conditions. The results revealed distinct bacterial community profiles between rice varieties under Cd stress. These findings contribute to understanding the complex interactions between rice plants, soil microbiota, and heavy metal dynamics.  

  • For a general reader, please write the complete term before using its abbreviation. Afterwards, write the abbreviation of that term concerned. Use this practice throughout the manuscript.
  • Please go through the whole manuscript and correct grammatical mistakes
  • Please improve the abstract
  • Please write keywords in alphabetical order
  • Add a detailed subsection to Materials and Methods describing the DNA extraction protocol, including kit name, modifications for soil samples, and quality control procedures.
  • Specify the 16S rRNA gene variable regions targeted (e.g., V3-V4 , V4-V5 ) and provide the exact primer sequences used.
  • Add details about the spatial distribution of the 10 soil samples taken within each treatment
  • Include information about which root zones were sampled (e.g., topsoil only, entire root system, specific root depth)
  • Add a brief discussion paragraph addressing how rhizosphere community dynamics might evolve throughout the rice growth cycle and their potential implications for the observed cadmium accumulation patterns
  • Add error bars (standard deviation or standard error) to all data points in figures where multiple replicates were analyzed
  • Please improve the discussion
Comments on the Quality of English Language
  • Please go through the whole manuscript and correct grammatical and typo mistakes

Author Response

Thanks for your kind suggestion. We have revised the document according to your review comments and highlighted the modified parts in green.

Point 1: For a general reader, please write the complete term before using its abbreviation. Afterwards, write the abbreviation of that term concerned. Use this practice throughout the manuscript.

Response 1: Thanks for your kind suggestion. We have written the complete term before using its abbreviations. Such as Cd(cadmium) in line 11, Yuzhenxiang (YZX) and Xiangwanxian 12 (XWX12) in line 74-75, soil organic matter (SOM), soil total nitrogen (TN), soil total phosphorus (TP), soil available nitrogen (AN), soil available phosphorus (AP), soil available potassium (AK) in line 98-100, Operational taxonomic units (OTUs) in line 114, A_1 (Low Cd_1, YZX), A_2 (Low Cd_2, XWX), B_1 (Medium Cd_2, YZX), B_2 (Medium Cd_2, XWX), C_1 (High Cd_3, YZX), and C_2 (High Cd_3, XWX) in line 126-127.

Point 2: Please go through the whole manuscript and correct grammatical mistakes

Response 2: Thanks for your kind suggestion. We have checked through the whole manuscript and corrected grammatical mistakes.

Point 3: Please improve the abstract

Response 3: Thanks for your kind suggestion. We have revised the abstract in line 10-24.

Point 4: Please write keywords in alphabetical order

Response 4: Thanks for your kind suggestion. We have reordered the keywords in alphabetical order.

Point 5: Add a detailed subsection to Materials and Methods describing the DNA extraction protocol, including kit name, modifications for soil samples, and quality control procedures.

Response 5: Thanks for your kind suggestion. We have redescribed the DNA extraction protocol in line 342-348.

Point 6: Specify the 16S rRNA gene variable regions targeted (e.g., V3-V4 , V4-V5 ) and provide the exact primer sequences used.

Response 6: Thanks for your kind suggestion. We have specified the 16S rRNA gene variable regions targeted and provide the exact primer sequences used in line 349-352.“The V3-V4 hypervariable regions of the bacterial 16S rRNA genes were amplified using the above DNA templates of the primers 338F (5’-ACTCCTACGGGAGGCAGCAG-3’) and 806R (5’-GGACTACHVGGGTWTCTAAT-3’)”

Point 7: Add details about the spatial distribution of the 10 soil samples taken within each treatment

Response 7: Thanks for your kind suggestion. We have added the details about the 30 soil samples in line 338-339.“Our experiments were divided into 6 groups (i.e., A_1, A_2, B_1, B_2, C_1, and C_2). Each group contained 5 biological replicates.”

Point 8: Include information about which root zones were sampled (e.g., topsoil only, entire root system, specific root depth)

Response 8: Thanks for your kind suggestion. We added the detailed information about root zones in line 340-342. “After the removal of foreign matter such as impurities, stones, and plant residues, we used a spoon to scrape the rhizosphere soil from the rice roots.”

Point 9: Add a brief discussion paragraph addressing how rhizosphere community dynamics might evolve throughout the rice growth cycle and their potential implications for the observed cadmium accumulation patterns

Response 9: Thanks for your kind suggestion. We have added a brief discussion paragraph addressing how rhizosphere community dynamics might evolve throughout the rice growth cycle and their potential implications for the observed cadmium accumulation patterns in line 269-276.

Point 10: Add error bars (standard deviation or standard error) to all data points in figures where multiple replicates were analyzed

Response 10: Thanks for your kind suggestion. We have added the error bars to all data points in figure1.

Point 11: Please improve the discussion

Response 11: Thanks for your kind suggestion. We have revised the discussion part in line 217-277.

Reviewer 2 Report

Comments and Suggestions for Authors

The present research tested two rice cultivars on three types of soils contaminated with Cadmium and evaluated the accumulation of Cd in the different tissues of the plants, as well as the modification on soil properties and bacteria distribution.

The introduction is globally well written with good references and present the research question clearly. But the first paragraph should be completely rewritten, reference 1 and 2 are not at all suitable according to the statement associated. Then the following sentence is too long and not clear with many cross information.

The results are well presented but the legend of the figures should be completed nowhere there is an explanation of which were the Cd modalities tested, or is said that we are comparing to different genotypes. Sometimes there is a kind of explanation of the results which is not the place. You could precise the number of repetitions used to make the mean and the error bars (which are not so clearly explained in the legends. THere are many abbreviation which are missing in the legends, but also the first time when cited in the text (all along the manuscript).

The discussion is complete with the associated good references.

Material and method should be completed with a more clear and detailed description of the experimental design, state more clear the number of repetition per trrrreatment and the disposition of the modalities, maybe it would be useful to have a schema of the distribution of the samples.

Then there are some missing data to be able to reproduce the method.

Conclusion and abstract are fine.

Please see more detailed comments in the attached document

Author Response

Thanks for your kind suggestion. We have revised the document according to your review comments and highlighted the modified parts in yellow.

Point 1: The introduction is globally well written with good references and present the research question clearly. But the first paragraph should be completely rewritten, reference 1 and 2 are not at all suitable according to the statement associated. (THese two references at not at all suitable for talking about the Rice market, as they focus on secondary metabolisms and transcriptomics respectively, it's not beacause they mention something about this in the introduction that means they are suitable. Here you should find some FAO publications or FAOstat data on which are the staple food in the world.)

Response 1: Thanks for your kind suggestion. We have replaced references 1 and 2 with some FAOstat data and the relevant reference in line 419-423.

Point 2: Then the following sentence is too long and not clear with many cross information. (the sentence is too long, with at least three main topics which are not completely correlated, please dissociate, the lack of suitable land to cultivate face to the increasing population, the origin of contamination of the soil with Cd and the impact on health of Cd. Besides it would be interesting to know how large is the area concerned by a contamination of Cd in the world.)

Response 2: Thanks for your kind suggestion.

We have completely rewritten the sentence in first paragraph in line 29-32.

“Annually, the lack of suitable land to cultivate faces the increasing population. Recently, some acreage of rice crops has been cultivated in contaminated soils due to multiple factors. The contaminated soil leads to the accumulation of multitudinous heavy metals in rice grains.”

Point 3: The results are well presented but the legend of the figures should be completed nowhere there is an explanation of which were the Cd modalities tested, or is said that we are comparing to different genotypes. Sometimes there is a kind of explanation of the results which is not the place. You could precise the number of repetitions used to make the mean and the error bars (which are not so clearly explained in the legends. THere are many abbreviations which are missing in the legends, but also the first time when cited in the text (all along the manuscript).

Response 3: Thanks for your kind suggestion. We have rewritten the legend of the figure1-figure 4. The details of revision are as following.

Figure 1. In line 92, we have replaced “Cd content in plant organs” to “the Cd content” .

In line 96, we have deleted the extra codon.

Figure 2. In line 105-107, we have rewritten the words to explanation with Cd concentration of the soil in the legend as “between the A (Low Cd_1, A)/B (Medium Cd_2, B) Cd content (A), A (Low Cd_1, A)/C (High Cd_3, C) Cd content (B), and B (Medium Cd_2, B)/C (High Cd_3, C) Cd content (C) treatments" .

Figure3. In line 123, we have indicated the meaning of the acronyms in the legend OTU (Operational taxonomic units). And added the means of A_1 (Low Cd_1, YZX), A_2 (Low Cd_2, XWX12), B_1 (Medium Cd_2, YZX), B_2 (Medium Cd_2, XWX12), C_1 (High Cd_3, YZX), and C_2 (High Cd_3, XWX) in line 126-137.

Figure 4. In line 160-161, we have indicated what A, BC, means as well know A-2, A1, C1 and so on. “Note: A_1 (Low Cd_1, YZX), A_2 (Low Cd_2, XWX12), B_1 (Medium Cd_2, YZX), B_2 (Medium Cd_2, XWX12), C_1 (High Cd_3, YZX), and C_2 (High Cd_3, XWX12).”

In line 186-190, we have revised the right Font Format of “)”.

In line 197, we have replaced the “abundant” to “abundance”.

We have added detailed information about the abbreviations when they first show up. Such as Cd(cadmium) in line 11, Yuzhenxiang (YZX) and Xiangwanxian 12 (XWX12) in line 74-75, soil organic matter (SOM), soil total nitrogen (TN), soil total phosphorus (TP), soil available nitrogen (AN), soil available phosphorus (AP), soil available potassium (AK) in line 98-100, Operational taxonomic units (OTUs) in line 114, A_1 (Low Cd_1, YZX), A_2 (Low Cd_2, XWX), B_1 (Medium Cd_2, YZX), B_2 (Medium Cd_2, XWX), C_1 (High Cd_3, YZX), and C_2 (High Cd_3, XWX) in line 126-127.

Point 4: Material and method should be completed with a more clear and detailed description of the experimental design, state more clear the number of repetition per trrrreatment and the disposition of the modalities, maybe it would be useful to have a schema of the distribution of the samples.

Response 4: Thanks for your kind suggestion. We have rewritten and completed with a more clear and detailed description of the experimental design as following.

It's not clear how the plants were distributed, how many plants were per repetition.What I understand, is two rows with 14 seedlings per genotype and repetition, where each plant is separated from the other 20cm in the row and between rows. Between Cd plot modalities there is and additional row?How many repetitions are per variety?How were distributed the modalities? ramdonly?

We have rewritten the sentence with detailed information in line 291-294.

“The experiment was arranged in a completely randomized block design with three replications. Each cultivar was grown in two rows of 14 hills per row, with two seed-lings transplanted per hill. Plant spacing was 17 cm × 20 cm (within rows × between rows)."

 “or” how you determined the one to the other?

We have replaced the “and” to “or” in line 295.

In line 294, not clear what you mean “management involves early flooding”? can you talk about mm of water? or something more precise?

We have rewritten the sentence with detailed information in line 296-301.

“Water management: intermittent irrigation with three controlled drought periods during the growth cycle: (i) peak tillering stage (3-day drought), (ii) post-heading stage (3-day drought), and (iii) pre-harvest stage (3-day drought). Continuous flooding was maintained during intervening periods. Irrigation was resumed upon visible initiation of soil surface cracking to establish standardized moisture stress for phenotypic selection.”

In line 302-303, “The soil properties and this,” shouldn't be in the next paragraphe?

We have removed the sentence in line 302-303 to line 319-320.

In line 304, “separately” do you mean, repetition per repetiion, plant by plant or each part (roots, leaves...) please be more clear what you did exactly

We replaced the “repetition per repetition” to “separately” in line 307.

In line 307,and nitric acid and hydrogen peroxide were added” how much of each per sample?

We have added the detailed information as “nitric acid (5ml) and hydrogen peroxide (2ml) were added” in line 310-311.

Round 2

Reviewer 2 Report

Comments and Suggestions for Authors

The authors have now provided a corrected version of the document, 

they have replaced the two references, they have completed the figures legends to make them understandable without the text. The material and method has also been completed and now we can reproduce the experimentation properly.

Besides they have made a lot of improvements in the discussion

Author Response

Thanks a lot for your sincerely and useful suggestions,which help us improved the manuscript.